# Corrosion Resistance of 316L/CuSn10 Multi-Material Manufactured by Powder Bed Fusion

**DOI:** 10.3390/ma15238373

**Published:** 2022-11-24

**Authors:** Robert Kremer, Johannes Etzkorn, Heinz Palkowski, Farzad Foadian

**Affiliations:** 1Faculty of Mechanical Engineering, Dortmund University of Applied Sciences and Arts, Sonnenstr. 96, 44139 Dortmund, Germany; 2Institute of Metallurgy, Clausthal University of Technology, Robert-Koch-Strasse 42, 38678 Clausthal-Zellerfeld, Germany

**Keywords:** additive manufacturing, powder bed fusion, multi-material components, corrosion, microstructure, optimisation, stress relief annealing

## Abstract

Research and industry are calling for additively manufactured multi-materials, as these are expected to create more efficient components, but there is a lack of information on corrosion resistance, especially since there is a risk of bimetallic corrosion with two metallic components. In this study, the corrosion behaviour of a multi-material made of 316L and CuSn10 is investigated before and after a stress relief annealing using linear sweep voltammetry. For this purpose, a compromise had to be found in the heat treatment parameters in order to be able to treat both materials together. In addition, additively manufactured and rolled samples were investigated and used as a reference. Interaction of the two materials in the multi-material could be demonstrated, but further investigations are necessary to clearly assess the behaviour. In particular, the transition region of the two materials should be investigated. In this study, a stress relief heat treatment at 400 °C caused a slight improvement in the corrosion resistance and reduced the scatter of the measurements significantly. No significant difference was measured between the additively produced and rolled samples.

## 1. Introduction

Research and industry demand for additively manufactured multi-materials, as these are intended to create more efficient components [1,2,3,4]. For instance, by combining 316L and CuSn10, 316L’s high mechanical–technological and corrosion-resistant properties can be combined with the good thermal conductivity of the copper-based alloy. This is a promising solution for, e.g., complex cooling channels in various applications [5,6]. However, joining materials with significantly different thermophysical properties is challenging and can lead to insufficient adhesion between materials during use [7]. Therefore, a sufficient diffusion layer between two materials is beneficial, as it creates a strong bonding and a smooth transition into each other. Powder Bed Fusion (PBF), which belongs to the Metal Additive Manufacturing (MAM) processes, is particularly suitable. In addition, the PBF process offers enormous freedom in design [8]. In this process, the powder is melted locally, layer by layer, with the help of a laser beam. Depending on the locally implemented energy, the melt pool generated by the laser beam can reach several layers, allowing the individual layers to grow together into a fully bonded part [9]. These deep melting traces lead to a mixing of the two materials, which is why a robust transition zone based on metallurgical bonds is created. [1]. However, the solid and direct connection of two materials with different positions in the electrochemical voltage series bears the risk of bimetallic corrosion [10]. The resulting galvanic element induces a current that damages the less noble material through electrochemical erosion [11]. To understand the corrosion behaviour of additive manufactured multi materials, it is important to know the behaviour of the individual materials. The literature already shows many papers on the corrosion of PBF-manufactured materials. In a review of the corrosion behaviour of additively manufactured metals. [12] It shows that additively manufactured materials are often less resistant to corrosion than conventionally manufactured materials. However, it is found that the many variables of the PBF process exert an influence on corrosion capability. In particular, rapid solidification and the formation of intermetallic phases are cited in this context. According to the review, there is a lack of holistic understanding [12,13,14]. Accordingly, the behaviour of PBF-manufactured components is difficult to predict, which explains divergent results in different studies. The literature also shows that heat treatments have an influence on corrosion resistance and can be used for material optimisation [13,14]. It should be noted that PBF-manufactured components are often heat-treated after production to reduce the residual stress introduced by the manufacturing process [15,16,17]. It is, therefore, necessary to know not only the corrosion behaviour in the as-built condition but also in the heat-treatment conditions. The research cited deals with individual materials and, accordingly, did not consider multi-materials. Similarly, standards and guidelines for PBF production deal exclusively with individual materials. Furthermore, multi-materials can only be heat-treated together, which usually means that a compromise has to be found in the choice of parameters, which can lead to deviations in the result.

Although corrosion is a ubiquitous challenge in the industry, research results in the field of corrosion susceptibility are still inconsistent. For example, there are no clear recommendations on how to investigate corrosion in PBF components. Accordingly, there are different investigations, which results in difficult comparability. Furthermore, there is only a little information on multi-materials made of PBF. This study contributes to research in this field.

In order to be able to reliably identify and classify the corrosion behaviour and the influence of the heat treatment, in this study, in addition to the multi-material, individual samples of 316L and CuSn were produced in the PBF process and examined together with purchased rolled samples as a reference. By generating its own reference values, the influence of different measurement and production parameters described in the literature is minimised. Accordingly, both 316L and CuSn samples were produced in this study using the PBF process, as well as multi-material samples. The single-material specimens serve as a reference for the multi-material specimens. The samples are examined in as-built and heat-treated conditions. Purchased rolled samples of 316L and CuSn were examined for better classification. Linear Sweep Voltammetry (LSV) was used as the investigation method, and hardness measurements and metallographic investigations were carried out for a better understanding. In addition to the main question, the experimental design can also answer other important questions. In particular, the following research questions on corrosion behaviour could be answered:Corrosion behaviour of the multi-material compared to single material.The influence of stress relief annealing on the multi-material and the single materials.The difference between rolled and additively produced material.

## 2. Materials and Methods

Gas-atomised CuSn10 and 316L powders from “m4p material solutions GmbH” (Austria) were used for manufacturing the PBF samples. SEM investigations were used to determine the particle size and morphology. They can be seen in Figure 1 and Figure 2, respectively. The average particle size was determined to be (23.7 ± 9.1) μm for 316L and (17.3 ± 7.3) μm for CuSn10. In addition, the flowability was investigated using flow analysis (Ø 5 mm), where flow times of (8.7 ± 0.3) s for 316L and (7.1 ± 0.3) s for CuSn10 were measured.

Sheets of dimensions 50 × 30 × 2 mm^3^ were used to study corrosion properties and microstructures. Sheets of 316L, CuSn10, and multi-material sheets were produced using PBF. The MLab R PBF system from Concept Laser was used for this purpose. The parameters used to produce the samples are listed in Table 1, and Figure 3 shows the scanning strategy used. In addition, the relative density, yield strength, and tensile strength are given, which were achieved during the parameter testing.

They were determined experimentally in advance, and the densities, which were measured during the tests, were also reported. During fabrication, all specimens were aligned with the long edge facing the build plate. The connection was made via grid support. For the multi-material specimens, the lower part (Z = 15 mm + support) was first made of 316L, and then the starting material was changed to manufacture the second half with CuSn10. The sequence was chosen because the energy input is higher with CuSn10 than with 316L. Thus, the energy transmitted during CuSn10 processing was sufficient to melt the already manufactured 316L and thus produce an adequate joint. In the reverse order, problems with material bonding are expected. 

The creation of the stress-relieved samples entailed heat treatment at a temperature of 400 °C for 20 min and subsequent cooling in air. As there is no standardised temperature for stress relief annealing for the manufactured multi-material, one had to be defined. The aim is to stress relieve both materials without causing recrystallisation. Since both materials differ in terms of the required temperatures (316L approx. 450–600 °C and CuSn approx. 250–400 °C) and holding times, a compromise had to be made. For the temperature, the bronze limits the maximum temperature, as recrystallisation starts earlier here than with 316L. The 316L was used as a guide for the holding time. As a compromise solution for both materials, 20 min at 400 °C annealings followed by cooling in air were defined. In addition to the PBF samples, conventionally manufactured 316L and CuSn6 sheets were used for comparison purposes. Due to supply difficulties, no conventionally produced CuSn10 was available, so CuSn6 was used instead. Due to the similar chemical composition and the identical phase, this was considered acceptable for comparison purposes. The eight investigated samples are listed in Table 2. The sample label indicates the material, production method, and heat treatment.

The chemical composition of all samples was investigated using Optical Emission Spectrometry (OES). The microstructure was determined by metallographic preparation. The steel was treated with a V2A etchant (50% HCl, 10% HNO_3_, 40% H_2_O) for etching at 70 °C. For the CuSn samples, different etchants had to be used. A solution of H_2_O_2_, NH_3_, and H_2_O was used for the additively produced CuSn samples, which had no effect on the rolled CuSn samples. These were then prepared with a solution of HCl, FeCl_3_, and H_2_O. It should be noted that for the CuSn-AM samples, the exposure time had to be increased for the annealed samples. For the hardness measurements, 20 points evenly distributed on the sample were measured using Vickers hardness HV10.

The corrosion behavior was investigated using Linear Sweep Voltammetry (LSV). For this purpose, all specimens were polished with 120-grit sandpaper to reduce the effect of roughness and to achieve better comparability. The corresponding measurements were carried out with the Galvanostat/Potentiostat Autolab PGSTAT204 from Metrohm. A three-electrode arrangement was used with a secondary calomel electrode as reference electrode and a platinum electrode as counter electrode. A circular sample surface of 1 cm^2^ was defined with a plating tape (PTC1, Gamry International, Warminster, PA, USA) and exposed to the electrolyte (NaCl, 3.5 wt%) at 35 °C in a thermostatically controlled cell. Prior to each LSV measurement, the open circuit potential (OCP) was recorded for 600 s to ensure stable conditions at the interface between the sample and electrolyte. According to the results of the preliminary experiments, a potential sweep from −0.5 to 0 V was performed against the reference electrode with a sampling rate of 5 mV/s. The measurement data were evaluated with Metrohm’s Nova software. For each sample type, six measurements were performed.

## 3. Results

The production of the measurement samples to be examined functioned without any malfunctions. Table 3 shows the measured chemical composition of the samples. The phosphorus content of the additively manufactured CuSn samples is strikingly high but corresponds to the specifications of the powder supplier. Otherwise, no abnormalities can be detected.

Figure 4 shows samples from the 316L-AM, CuSn-AM, and 316L/CuSn series before and after the LSV measurement without the masking tape. The traces of the test can be clearly seen on the CuSn in the measuring area. This has clearly darkened in the measuring range, while only about pitting is visible on the 316L. The microstructure in Figure 5 shows no differences between the annealed and untreated PBF samples. The melting traces are generally visible, and a fine microstructure can be seen. The rolled samples exhibit a microstructure typical for this process. Compared to the additively manufactured ones, they are significantly coarser. It was striking that the acid of water, ammonia, and hydrogen peroxide used for the additively manufactured CuSn samples showed no effect on the rolled samples. The rolled CuSn samples were, therefore, treated with a solution of water, ferric chloride, and hydrochloric acid. For the stress-relieved samples, the exposure time of the acid had to be increased from 20 to 30 s. The 316L samples could be etched identically.

Figure 6 shows the measured hardness of the samples. The hardness of the 316L and CuSn part of the multi-material sample corresponds to the hardness of the individual materials. Stress relief annealing had no decisive influence on the samples tested. Table 4 shows the measured resting potentials *OCP*, corrosion potentials *E_cor_,* and corrosion current density *I_cor_*. 

In the general evaluation of electrochemical measurements, the corrosion resistance of a material increases with increasing (more positive) values of the *OCP* and the corrosion potential *E_cor_* and with decreasing corrosion current density *I_cor_*.

## 4. Discussion

The collected data will be analysed in a targeted manner and correlated with each other to be able to investigate the questions posed at the beginning.

Difference between the multi-material and the single-material sample.

The heat-treated samples with their lower standard deviation were used for investigation. For the evaluation of the hardness measurements, the values of the individual samples (316L-AM-A and CuSn-AM-A) were compared with the respective parts of the 316l/CuSn-AM-A samples. No significant difference could be detected. The corresponding LSV data are presented in Table 5 and plotted in Figure 7. It can be seen that the characteristic values of the multi-material sample are not equal to the mean value of the two individual samples. Accordingly, an interaction between 316L and CuSn in the multi-material can be assumed. The resting potential *OCP* of the multi-material is comparable to that of the CuSn and clearly more negative than that of the 316L. The corrosion stress *E_cor_* of the multi-material is in the order of magnitude of that of CuSn and is clearly more positive than that of 316L. The multi-material shows the lowest value for the corrosion current *I_cor_*. About half as much as the 316L and many times less than the CuSn. The comparatively negative *OCP* of the multi-material and the comparatively less negative *E_cor_* show a contradictory picture since stronger negativity is associated with a higher susceptibility to corrosion. It is assumed that this apparent contradiction is associated with the chosen measurement method, but this could not be conclusively confirmed. In order to be able to make a statement about the corrosion behaviour, the corrosion current density is used as a measure since it has the closest correlation to the removal rate. As a result, the multi-material sample has the lowest corrosion current density and can be considered corrosion-resistant. To verify this statement, the surfaces of the LSV measuring areas were examined for corrosion damage.

Figure 8 shows representative areas of the samples attacked by corrosion at 200× magnification. The upper images show the difference between the single sample 316L-AM-A (a) and the steel of the multi-material sample 316L/CuSn-AM-A (b). It can be seen that the single material shows a stronger attack by pitting corrosion. More and deeper holes can be seen over the entire surface. (c) and (b) compared to the CuSn material according to the same scheme. The single sample shows more isolated pitting which is not found on the bronze part of the multi-material sample. However, both bronze surfaces show traces of uniform surface corrosion. 

At 2500× magnification under the SEM, see Figure 9, no difference can be seen between 316L-AM-A (a) and the steel of the multi-material (b) away from the pitting. Accordingly, only pitting corrosion is present. The bronze material shows a clear difference between the single material of the CuSn-Am-A sample (c) and the bronze of the multi-material (d). At high magnification, the two-dimensional track by the corrosion becomes visible. This is stronger and more uneven in the single material than in the multi-material.

Since the optical evaluation of the surfaces affected by corrosion shows a lower and more even wear of the multi-material compared to the individual materials, the decision to use the *I_cor_* to evaluate the susceptibility to corrosion is confirmed. Accordingly, the multi-material is considered to be more resistant to corrosion. This is due to the interaction between the two materials that was demonstrated at the beginning. The fear expressed that the connection of two materials with different positions in the stress series poses a risk to the corrosion resistance could not be proven here. Even more, the LSV measurement carried out indicates improved corrosion protection. The reason for this could be that the two materials do not lie next to each other but rather merge into each other. In other studies, it was shown that during the formation of the transition, newly melted material is drawn to the bottom of the molten bath by the Marangoni flow, resulting in a flowing transition. [2] In [1], a transition area of 600 µm could be detected, which could be divided into three areas: one FE- and one Cu-dominated area each, as well as a transition area in between, which could not be clearly assigned to a Cu- or Fe-matrix. These areas are interspersed with local material accumulations in which the material did not dissolve. This is due to the high cooling rate of up to 108 K/s of the manufacturing process. [2] It is assumed that the higher corrosion resistance in the measuring range of the multi-material is due to this complex transition area, which is why it must be focused on in subsequent work.

Influence of stress relief annealing.

In the metallographic micrographs (see Figure 5) as well as in the hardness measurements (see Figure 6), no difference was found before and after stress relief heat treatment. The results of the corresponding LSV measurements are listed in Table 6 and plotted in Figure 10. The rest potential shows no significant changes due to the heat treatment, while the corrosion potential becomes more positive. According to the more positive corrosion potential and the lower corrosion current density after the heat treatment of the 316L, CuSn, and multi-material samples, an improvement of the corrosion resistance by stress relief annealing can be assumed. In addition, the scatter of the LSV measurements decreased significantly after heat treatment, which indicates a homogenisation in the microstructure.

In investigation of Kong et al. [18], no significant change in the corrosion behaviour of PBF-manufactured 316L could be brought about by stress-relief heat treatment. In work done by Zhou et al. [19], 316L was stress-relief annealed at 500 °C after PBF fabrication, which probably resulted in the extinction of dislocations. However, this reduction in residual stresses was not found to be significant for corrosion behaviour. In the electrochemical measurements, the measured passive current and the corrosion current showed slight improvements. In both studies, the temperatures during stress relief annealing were considered too low to significantly reduce the corrosion behaviour. Zeng et al. [20] have reported that annealing at 600 °C for CuSn leads to improved corrosion resistance and larger grains. Furthermore, an XR measurement shows that after annealing, only al-pha-Cu(Sn) is present, while alpha and delta (Cu41Sn10) are found before annealing. It is assumed that these changes also occur during stress-relief annealing. In the studies cited, the measurement and production parameters differ, which makes comparison difficult.

Similar behaviour is described in the literature for the influence of heat treatment. However, the comparison is complicated by the different conditions. The reduced residual stresses after heat treatment are not described in the literature. However, it is assumed that these are due to relaxation effects in the material. There are no comparative values available for the multi-material, but the picture is expected when considering the individual materials. Since, according to the literature, a significant change in 316L only occurs at higher temperatures, and it could be shown that the CuSn alloy could be improved positively in terms of corrosion behaviour at 600 °C, an annealing temperature of 600 °C is recommended for multi-materials made of 316L and CuSn. However, the mechanical-technological properties still have to be tested.

Difference between rolled and additively produced material.

For evaluation, the additively manufactured samples 316-AM and CuSn-AM are compared with the respective rolled reference materials 316L-R and CuSn-R. A clear difference in the microstructure between the PBF and the rolled samples (see Figure 5) is stated, which is not reflected in the corrosion tests. The results of the LSV measurement in Table 7 and Figure 11 show the resting potential, corrosion potential, and corrosion current density of the corresponding 316L and CuSn samples. The characteristic values of the LSV measurements do not show any significant difference between rolled and additively produced samples, despite a clearly different microstructure, which leads to the expectation of comparable corrosion behaviour. However, the surface of the examined 316L area shows more pitting in the case of rolled samples. In other work, less manganese sulphide was found in PBF-manufactured 316L than in rolled material, which can be attributed to the rapid solidification of the melt in the PBF manufacturing process. Since manganese sulphide favours pitting corrosion [21], the better resistance of 316L-AM can be attributed to a likely reduced amount of manganese sulphide [22]. Furthermore, the CuSn-AM and CuSn-R samples could not be prepared with the same etchant.

## 5. Conclusions

No increased hazard due to the interaction of the two materials of the multi-material could be demonstrated in this study. Furthermore, a heat treatment for stress-relief annealing, which is common after PBF production, can be regarded as uncritical and has a positive influence on corrosion stability. It needs to be further investigated how to design the parameters for joint heat treatment.In the case of the multi-material, an interaction between the two materials could be demonstrated. The LSV measurements show partially contradictory results. However, the corrosion current and visual observation of the corroded surfaces indicate a higher corrosion resistance of the multi-material compared to the single material. The complex transition between the two materials is assumed to be the cause, which must be focused on in further investigations.The additively manufactured and the rolled samples show a different microstructure. The AM specimens have long, narrow grains that go along the build-up direction, while the rolled specimens have more round grains. Nevertheless, the corrosion behaviour did not differ significantly in the LSV investigation. However, a lower pitting tendency was observed in the additively manufactured samples, which could be attributed to a lower proportion of manganese sulphied.Stress-relief annealing at 400 °C for 20 min of the additively manufactured 316L, CuSn, and multi-material samples improved the corrosion resistance and reduced the scattering on the samples. However, the change in 316L is only slight. Supported by literature references, heat treatment of multi-material of 316L and CuSn at 600 °C is recommended for investigation.There is a need for a uniform specification of how PBF materials are to be tested for corrosion resistance, as the different test parameters in the literature make comparison considerably more difficult.

## Figures and Tables

**Figure 1 materials-15-08373-f001:**
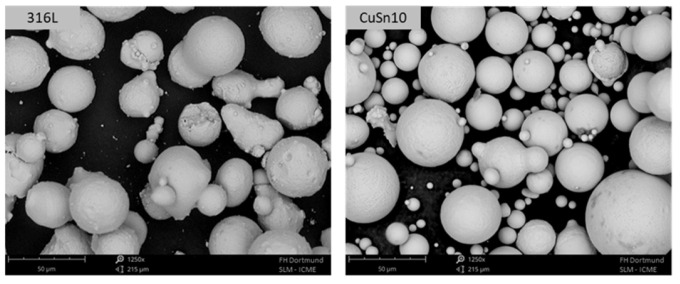
SEM images of the as-received powders 316L and CuSn10.

**Figure 2 materials-15-08373-f002:**
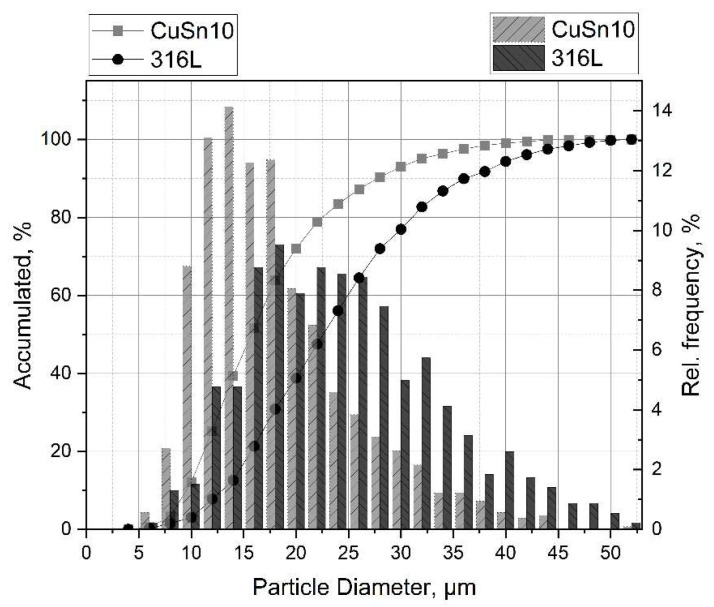
Distribution of powder particle sizes.

**Figure 3 materials-15-08373-f003:**
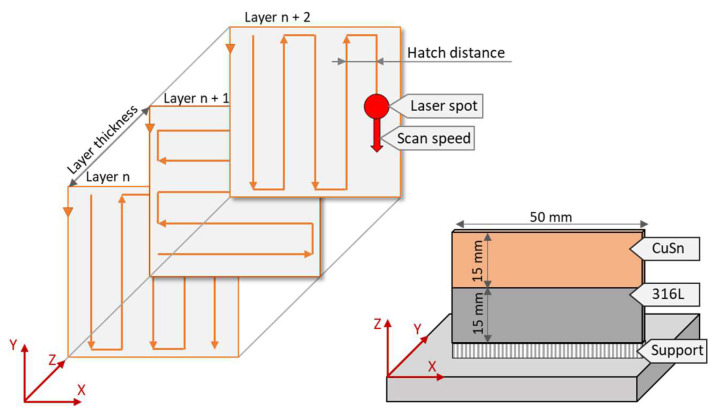
Schematic representation of hatching and path planning and the general fabrication of the multi-material.

**Figure 4 materials-15-08373-f004:**
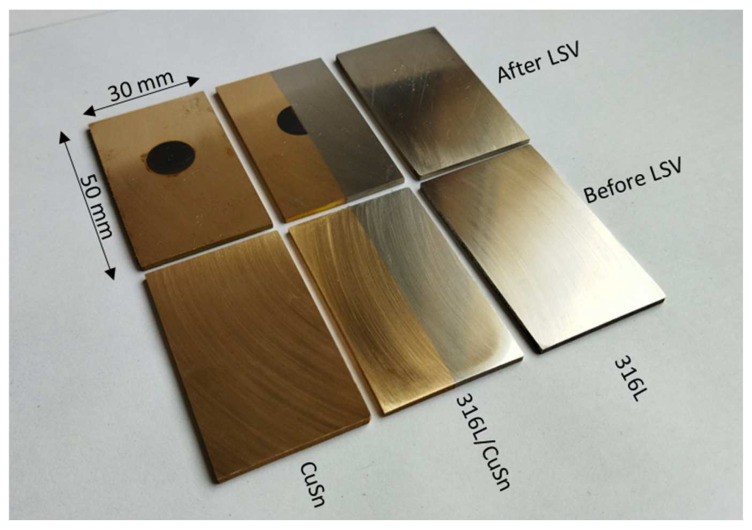
Manufactured and post-processed AM samples before and after an LSV measurement.

**Figure 5 materials-15-08373-f005:**
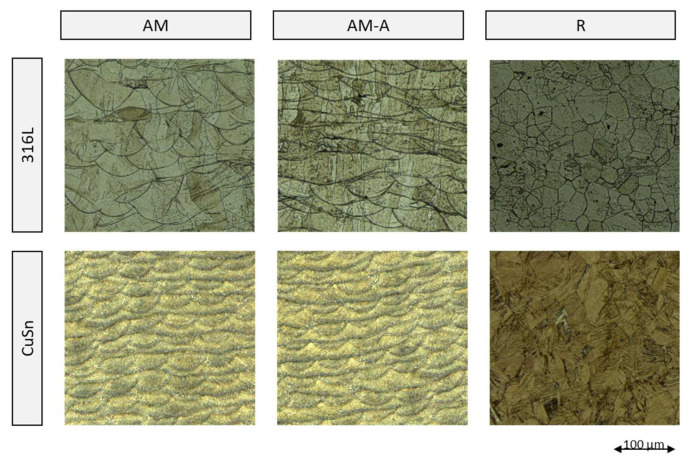
Comparison of investigated microstructures.

**Figure 6 materials-15-08373-f006:**
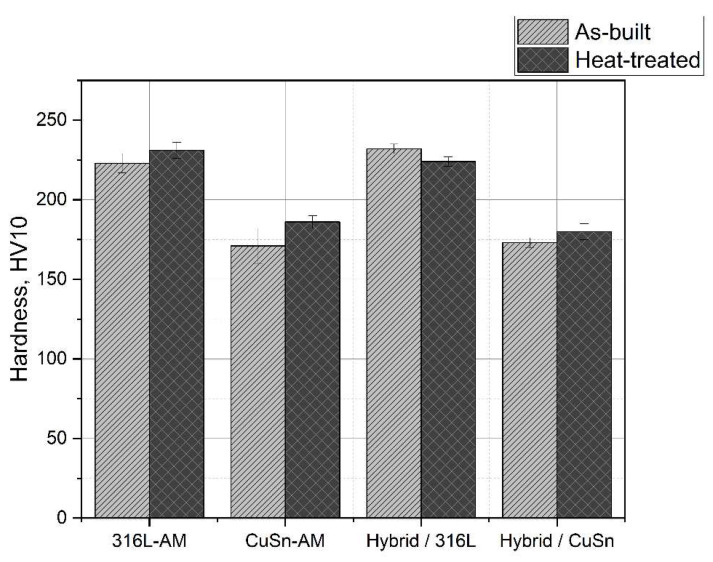
Overview of the hardness values.

**Figure 7 materials-15-08373-f007:**
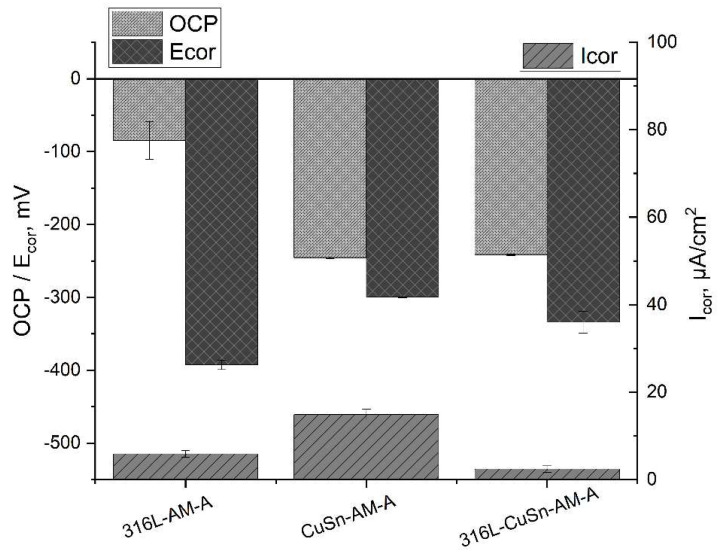
Comparison of the single and multi-materials.

**Figure 8 materials-15-08373-f008:**
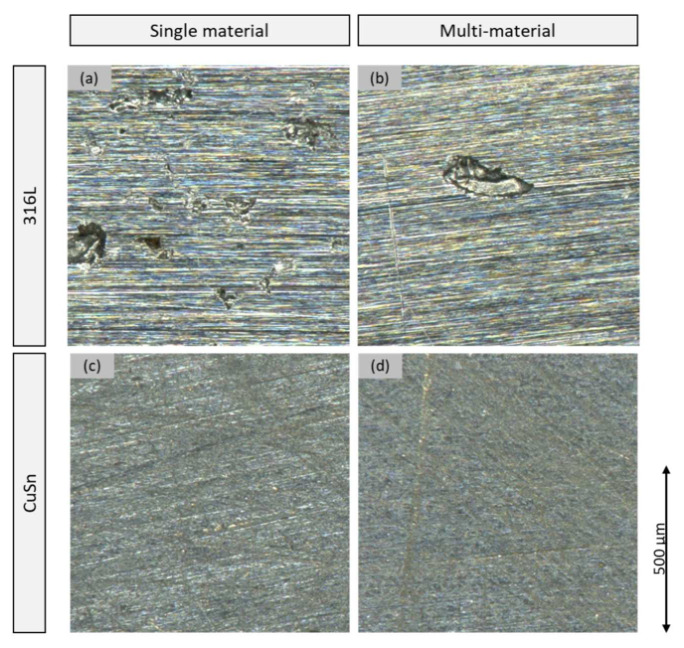
Surfaces after the LSV test under the light microscope (**a**) 316L-AM-A; (**b**) 316L/CuSn-AM steel side; (**c**) CuSn-AM-A; (**d**) 316L/CuSn-AM bronze side.

**Figure 9 materials-15-08373-f009:**
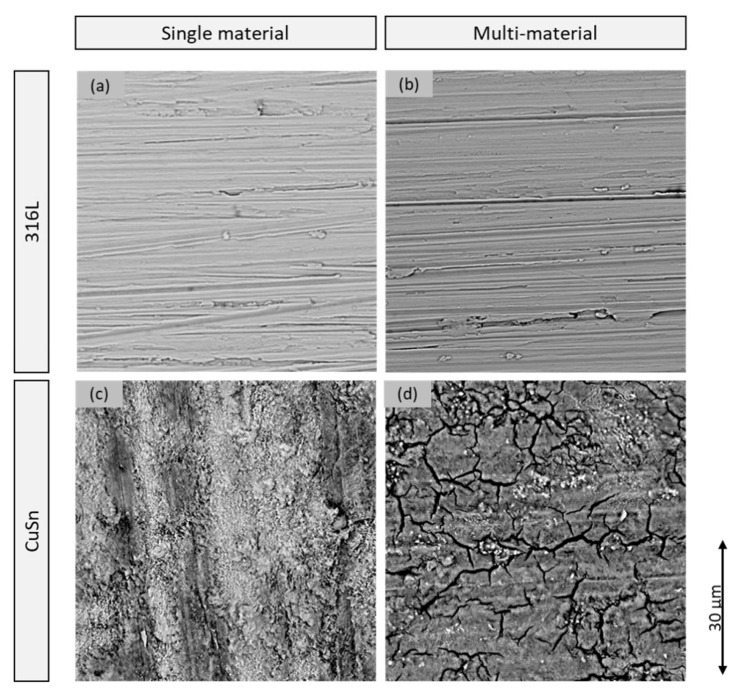
Surfaces after the LSV test under the scanning electron microscope (**a**) 316L-AM-A; (**b**) 316L/CuSn-AM steel side; (**c**) CuSn-AM-A; (**d**) 316L/CuSn-AM bronze side.

**Figure 10 materials-15-08373-f010:**
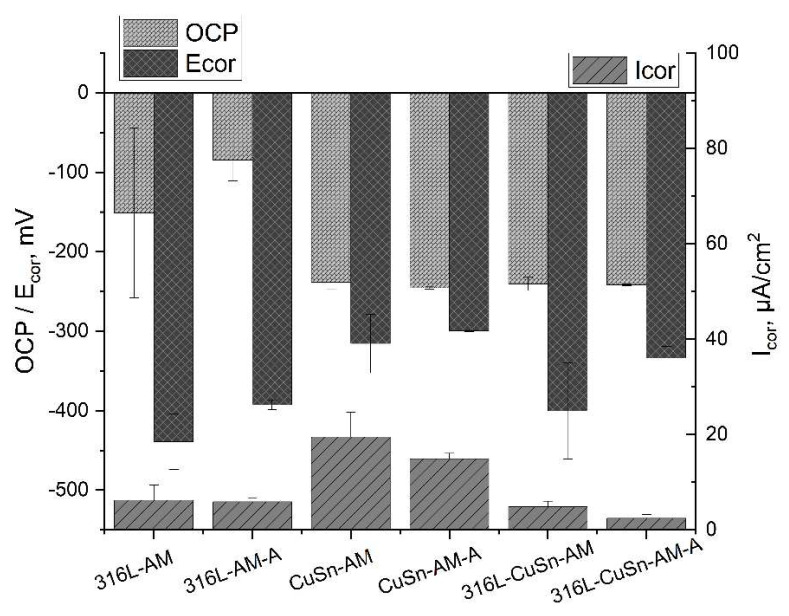
Comparison of the additive samples before and after heat treatment.

**Figure 11 materials-15-08373-f011:**
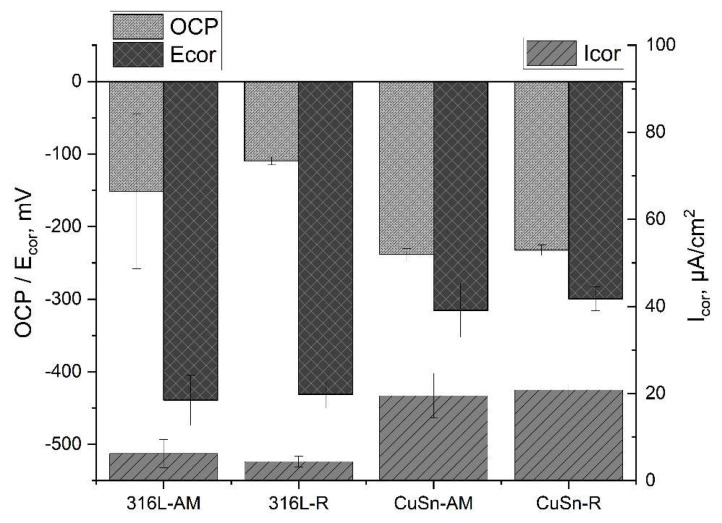
Comparison of the LSV results of additively produced samples with the rolled samples.

**Table 1 materials-15-08373-t001:** Parameters used for manufacturing the samples.

Parameter	316L	CuSn10
Laser power	90 W	100 W
Scan speed	800 mm/s	324 mm/s
Hatching	0.070 mm	0.065 mm
Layer thickness	0.025 mm	0.015 mm
Inert gas	Nitrogen	
Relative density	99.95%	99.7%
Yield Strength	573 ± 7 MPa	420 ± 14 MPa
Ultimate Strength	632 ± 10 MPa	487 ± 12 MPa

**Table 2 materials-15-08373-t002:** List of investigated samples.

Designation	Description
316L-AM ^1^	Additively manufactured 316L
316L-AM-A ^2^	Additively manufactured and stress-relieved 316L
CuSn-AM	Additively manufactured CuSn10
CuSn-AM-A	Additively manufactured and stress-relieved CuSn
316L/CuSn-AM	Additively manufactured multi-material
316L/CuSn-AM-A	Additively manufactured and stress-relieved multi-material
316L-R ^3^	Roled 316L
CuSn-R	Roled CuSn6

^1^ Additive manufacturing; ^2^ Stress relieved; ^3^ Rolled.

**Table 3 materials-15-08373-t003:** Chemical composition of the samples.

Element	316L-AM	316L-R	CuSn-AM	CuSn-R
Cu [%]	0.048 ± 0.075	0.336 ± 0.0098	88.81 ± 0.068	94.21 ± 0.0065
Sn [%]	-	-	10.72 ± 0.064	5.67 ± 0.118
P [%]	0.026 ± 0.001	0.035 ± 0.0009	0.368 ± 0.0034	0.078 ± 0.530
Fe [%]	68.11 ± 0.08	68.80 ± 0.100	0.014 ± 0.0096	<0.001
C [%]	0.025 ± 0.01	0.018 ± 0.0005	-	-
Mn [%]	1.26 ± 0.01	1.17 ± 0.0080	<0.0005	<0.0005
Si [%]	0.78 ± 0.01	0.368 ± 0.0034	<0.001	<0.001
S [%]	0.013 ± 0.000	0.0090 ± 0.0002	0.0032 ± 0.0001	0.0022 ± 0.0001
Cr [%]	16.37 ± 0.04	16.86 ± 0.190	-	-
Ni [%]	10.81 ± 0.01	9.92 ± 0.054	0.028 ± 0.002	0.0082 ± 0.0002
Mo [%]	2.61 ± 0.01	2.08 ± 0.018	-	-

**Table 4 materials-15-08373-t004:** Determined characteristic values of the LSV investigations.

Sample	*OCP* [mV]	*E_cor_* [mV]	*I_cor_* [µA/cm^2^]
316L-AM	−151 ± 107	−439 ± 35	6.21 ± 3.21
316L-AM-A	−85 ± 26	−393 ± 6	5.87 ± 0.75
316L-R	−110 ± 5	−431 ± 19	4.35 ± 1.23
CuSn-AM	−239 ± 8	−316 ± 37	19.49 ± 5.14
CuSn-AM-A	−246 ± 1	−299 ± 1	14.91 ± 1.14
CuSn-R	−233 ± 7	−300 ± 16	20.81 ± 1.49
316L/CuSn-AM	−241 ± 8	−400 ± 61	4.88 ± 1.11
316L/CuSn-AM-A	−242 ± 1	−334 ± 15	2.43 ± 0.82

**Table 5 materials-15-08373-t005:** LSV characteristic values of single and multi-materials.

Sample	*OCP* [mV]	*E_cor_* [mV]	*I_cor_* [µA/cm^2^]
316L-AM-A	−85 ± 26	−393 ± 6	5.87 ± 0.75
CuSn-AM-A	−246 ± 1	−299 ± 1	14.91 ± 1.14
316L/CuSn-AM-A	−242 ± 1	−334 ± 15	2.43 ± 0.82

**Table 6 materials-15-08373-t006:** LSV measured values of the additive samples before and after heat treatment.

Sample	*OCP* [mV]	*E_cor_* [mV]	*I_cor_* [µA/cm^2^]
316L-AM	−151 ± 107	−439 ± 35	6.21 ± 3.21
316L-AM-A	−85 ± 26	−393 ± 6	5.87 ± 0.75
CuSn-AM	−239 ± 8	−316 ± 37	19.49 ± 5.14
CuSn-AM-A	−246 ± 1	−299 ± 1	14.91 ± 1.14
316L/CuSn-AM	−241 ± 8	−400 ± 61	4.88 ± 1.11
316L/CuSn-AM-A	−242 ± 1	−334 ± 15	2.43 ± 0.82

**Table 7 materials-15-08373-t007:** LSV measured values of the additively manufactured samples with the rolled samples.

Sample	*OCP* [mV]	*E_cor_* [mV]	*I_cor_* [µA/cm^2^]
316L-AM	−151 ± 107	−439 ± 35	6.21 ± 3.21
316L-R	−110 ± 5	−431 ± 19	4.35 ± 1.23
CuSn-AM	−239 ± 8	−316 ± 37	19.49 ± 5.14
CuSn-R	−233 ± 7	−300 ± 16	20.81 ± 1.49

## Data Availability

Not applicable.

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
