# Peer review of "Corrosion Resistance of 316L/CuSn10 Multi-Material Manufactured by Powder Bed Fusion"

_materials, 2022, doi:10.3390/ma15238373_

Round 1

Reviewer 1 Report

·       Throughout the entire manuscript, follow the AM terminology of the ASTM F3187 and ISO/ASTM  52900. For instance, change SLM (a trade market name) to PBF.

·       Line 41. Please, better explain the terminology "elastic bond". Usually, in the laser-based AM processes, the bond between the layers occurred via a metallurgical bond (i.e., solidification and crystallographic continuity between the layers).

·       Verify throughout the entire manuscript some typing errors.

·       Add a schematic representation of the specimen fabrication and the hatch and path planning of the deposition, as indicated in 2nd paragraph of the section 2.

·       This is a critical point of the manuscript that must be clarified. Please, clarified the reasons to choose the HT temperature and time. Is it based in some procedure/standard? ASTM F3184 – 16 (Standard Specification for Additive Manufacturing Stainless Steel Alloy (UNS S31603) with Powder Bed Fusion) did not recommend the HT selected. Also, to the bimetallic material (316L + CuSn10) is possible to select some HT that can simultaneously meet the requirement of the two materials or is necessary a development of a specific HT for bimetallic materials? These are key questions to understand the manuscript proposal.

·       The first phrase of the first paragraph of section 3 is disconnected from the remaining paragraph.

·       The different corrosion effects on the samples can be seen clearly”. Clearly appointed in fig 3 the difference between the conditions.

·       The metallography details must be in the materials and procedure section.

·       Last paragraph of section 3. To Indicate if the data of Table 3 is good or bad in a final user view, if the HT improves or decreases the corrosion resistance, and indicate if the bi-metallic component is an interest opportune from the corrosion resistance point of view. In addition, also make a previous explanation of what each result from corrosion tests means for a better comprehension of the reader, e.g., the increase of OCP is good or bad?

·       The authors must appoint some reasons why AM had higher pitting corrosion.

·       The authors must try to explain corrosion results and the effect of HT, not only describing the results and stating some conclusions, especially in the topic "Difference between the multi-material and the single material sample" the key point of the manuscript.

·       The conclusion of the"           Difference between the multi-material and the single material sample" was not supported by the Table 6 results. Please verified this.

·       The conclusion section must be focused on the manuscript objectives. So, the 1st conclusion must not have etching procedures.

Author Response

Dear Editor,

Thank you for giving us the opportunity to submit a revised draft of our manuscript to the Journal of Materials. We highly appreciate all the constructive and helpful comments and suggestions from the reviewers.

We thoroughly revised the manuscript, and we are pleased to incorporate almost all the points raised by the reviewers. Following is a point-by-point response to the reviewers. All the changes are highlighted within the manuscript. If you have any further questions, please do not hesitate to contact us.

Sincerely yours

Heinz Palkowski

Reviewer 2 Report

This paper compares the corrosion resistance and microstructure changes of selective laser melting CuSn10, 316L, multi-material parts and rolled materials before and after heat treatment, but there are some problems in this paper that need to be modified according to the following comments:

1.        The title of the article and the content of the study are inconsistent. Materials from different manufacturing processes were compared in the study, and only materials for additive manufacturing were mentioned in the title.

2.        Multi material components are often subject to electrochemical corrosion in nature. In the experiment, multi material components seem to be subject to electrochemical corrosion.

3.        The article does not analyze the current status of domestic and international research in this research direction, and cannot highlight the novelty of the research content of the article.

4.        In the article, only the surface microstructure and hardness of the sample after heat treatment are measured with few parameters, and other mechanical properties or parameters measuring surface quality such as strength, surface roughness, porosity, etc. are not measured.

5.        The article has some formatting issues and errors in the content section.

6.        The following is a recently published paper that mentions the method of characterizing the surface cleanliness, which may be helpful for the part of this paper on the cleanliness evaluation of surface after polishing. Please cite in the article if it is helpful, otherwise ignore it. “Effect of surface cleaning on interface bonding performance for 316H stainless steel joints manufactured by additive forging. Materials & Design, 2021, 210: 110025.”

Author Response

(The authors gave the same response as above.)

Round 2

Reviewer 1 Report

The authors clarified all the reviewer's comments.

Author Response

Thank you so much for your review.

Reviewer 2 Report

The author has revised and improved the reviewers' comments, explained the questions raised by the reviewers in detail, and the overall article has been significantly improved, but the following issues still need to be slightly repaired:

1.      In nature, due to the composition of the galvanic cell structure, multi-materials are often prone to electrochemical corrosion. But the experiment carried out in this paper is electrolytic corrosion.

2.      In Figure 5, there are slight differences in the microstructure of annealed and untreated samples, and more detailed microstructure of annealed samples.

3.      There are still some formatting problems in the article, such as: the table should be placed in the center, the table 1 should be 0.070 mm, the chemical formula subscript, etc.

Author Response

Dear Sir or Madam,

Thank you very much for your review. Below you will find our response (in red).

  1.   In nature, due to the composition of the galvanic cell structure, multi-materials are often prone to electrochemical corrosion. But the experiment carried out in this paper is electrolytic corrosion. In this work, we first analyzed electrolytic corrosion. Further investigations are planned for the future.  

2.      In Figure 5, there are slight differences in the microstructure of annealed and untreated samples, and more detailed microstructure of annealed samples.   

The slight differences are due to a small difference in the effect of the etchant. This leads to a lightly different visibility of the fusion traces. However, the differences in microstructure are so close together and within a usual preparation variation that we do not assume a significant deviation in this case.

3.      There are still some formatting problems in the article, such as: the table should be placed in the center, the table 1 should be 0.070 mm, the chemical formula subscript, etc.

We have revised the manuscript again and corrected some formatting issues. Since we used the journal's draft, some tables are formatted based on the layout. The format of the paper will certainly be checked and edited by the journal if the paper is accepted for publication.